# ANRIL regulates multiple molecules of pathogenetic significance in diabetic nephropathy

Parisa Sooshtari[1,2], Biao Feng[1], Saumik Biswas[1], Michael Levy[1], Hanxin Lin[1,3], Zhaoliang Su[4], Subrata Chakrabarti[1,3]*

**1** Department of Pathology and Laboratory Medicine, Western University, London, Ontario, Canada, **2** Ontario Institute for Cancer Research, Toronto, Ontario, Canada, **3** Department of Pathology and Laboratory Medicine London Health Sciences Centre, London, Ontario, Canada, **4** Department of Immunology Jiangsu University, Jiangsu, PR China

* subrata.chakrabarti@lhsc.on.ca

## Abstract

### Background

Hyperglycemia-induced transcriptional alterations lead to aberrant synthesis of a large number of pathogenetic molecules leading to functional and structural damage to multiple end organs including the kidneys. Diabetic nephropathy (DN) remains a major cause of end stage renal disease. Multiple epigenetic mechanisms, including alteration of long non-coding RNAs (lncRNAs) may play a significant role mediating the cellular transcriptional activities. We have previously shown that lncRNA ANRIL may mediate diabetes associated molecular, functional and structural abnormalities in DN. Here we explored downstream mechanisms of ANRIL alteration in DN.

### Methods

We used renal cortical tissues from ANRIL knockout (KO) mice and wild type (WT) mice, with or without streptozotocin (STZ) induced diabetes for RNA sequencing. The differentially expressed genes were identified using edgeR and DESeq2 computational methods. KEGG and Reactome pathway analyses and network analyses using STRING and IPA were subsequently performed.

### Results

Diabetic animals showed hyperglycemia, reduced body weight gain, polyuria and increased urinary albumin. Both albuminuria and polyuria were corrected in the KO diabetic mice. RNA analyses showed Diabetes induced alterations of a large number of transcripts in the wild type (WT) animals. ANRIL knockout (KO) prevented a large number of such alterations. The altered transcripts include metabolic pathways, apoptosis, extracellular matrix protein synthesis and degradation, NFKB related pathways, AGE-RAGE interaction pathways etc. ANRIL KO prevented majority of these pathways.

**Data Availability Statement:** Data is available from https://www.ncbi.nlm.nih.gov/geo/query/acc.cgi?acc=GSE197699.

**Funding:** Dr. Subrata Chakrabarti receiving funding from CIHR (169650), and from Jiangsu Province collaborative research Program (BX2019100). Dr. Parisa Sooshtari received funding from Lawson Health Research Institute and Ontario Institute for Cancer Research. Dr. Zhaoliang Su received funding from Jiangsu Province collaborative research Program (BX2019100).

**Competing interests:** The authors have declared that no competing interests exist.

## Conclusion

These findings suggest that as ANRIL regulates a large number of molecules of pathogenetic significance, it may potentially be a drug target for DN and other chronic diabetic complications.

## Introduction

Nephropathy is one of the debilitating chronic complications of diabetes. Diabetic nephropathy remains a major cause of end stage renal disease (ESRD) [1]. Starting with glomerular hyperfiltration there is progressive renal damage, manifesting as albuminuria, declining glomerular filtration rate [GFR], and ultimately ESRD. At the microscopic level the kidneys show glomerular hypertrophy, glomerulosclerosis, tubulointerstitial inflammation and fibrosis. Although over the last few years various novel therapies have been developed, a large number of diabetics still progress towards ESRD [2]. Hence better understanding of the pathogenesis and further delineation of pathogenetic mechanisms are needed to develop biomarkers for early diagnosis and novel treatment targets.

Hyperglycemia has established itself as the key factor for the initiation and progression of all chronic diabetic complications including DN [3]. However, in a tissue specific way the effect of hyperglycemia may vary, causing alteration of specific gene expression and ultimately causing tissue damage [4, 5]. Multiple mechanisms contribute to the initiation and progression of DN. Hyperglycemia leads to multiple cellular structural and functional changes in the organs. These include hemodynamic changes and cellular metabolic changes leading to altered gene transcription of several such vasoactive and growth factors such as VEGF, TGF β, setting the stage for tissue damage and organ failure [5]. Microscopically, kidneys demonstrate glomerular basement membrane thickening, increased mesangial matrix deposition, eventually manifested as glomerulosclerosis and interstitial fibrosis [6]. At the biochemical level, there is deposition and reduced degradation of extracellular matrix (ECM) proteins such as fibronectin [FN], type IV collagen (Col1α4) etc [6]. Functionally, there is increased urinary albumin excretion ultimately leading to renal failure [2, 3].

Epigenetic alterations regulating the synthetic process of macromolecules possibly plays a major role in all chronic diseases including DN [7, 8]. Such modifications include DNA methylation, histone modifications through acetylation and methylation and alterations of noncoding RNAs (ncRNAs) [9]. ncRNAs consists of several classes of RNA molecules including microRNAs and long noncoding RNAs (lncRNAs). Of particular relevance to this research, lncRNAs (200-2000nt) lack protein-coding capacity and are capable of regulating gene expression at both transcriptional and translational levels [10]. They can regulate local(*cis*) and distal [*trans]* genes by a large number of mechanisms and play key regulatory roles in diverse biological processes [10, 11].

lncRNA ANRIL, situated in the antisense direction of p15/CDKN2B-p16/CDKN2A-p14/ARF (*INK4b-ARF-INK4a*) gene cluster, has 19 exons and spans126.3kb [11]. We have recently reported that ANRIL is upregulated and plays a pathogenetic roles in several diabetic complications including DN [12, 13]. We have further shown that it works through an interaction with EZH2 of polycomb repressive complex 2(PRC2) and histone acetylator, p300 [12, 13]. ANRIL locus is a hotspot for multiple disease-associated polymorphisms and DNA alterations and has been steadily associated with cardiovascular diseases, cancer, diabetes, glaucoma and other conditions [14]. We have also demonstrated oxidative DNA damage in diabetes is a

potential mechanism of ANRIL upregulation in diabetes [15, 16]. In our previous study, we have shown that ANRIL blockade prevents diabetes-associated molecular, functional and structural abnormalities in the context of DN [13]. To explore the mechanism of ANRIL mediated prevention of several changes in DN at the transcriptional level, we carried out this research. We were specifically interested in ANRIL mediated transcriptional alterations in the context of DN.

Hence. to gain a better understanding of the downstream mechanisms and pathways of action of ANRIL and to identify potential drug targets, we took RNA-sequencing and bioinformatics analysis-based approach. To this extent, we analyzed renal cortical tissues of a type 1 model of diabetes, with or without ANRIL knockout.

## Methods

### Animals

All animals were cared for according to the guiding principles in the care and use of animals. The experiments were approved by Western University Animal Care and Veterinary Services. All experiments conform to the guide for care and use of laboratory animals published by the NIH (NIH publication no. 85–23, revised in 1996).

ANRIL knockout (129S6/SvEvTac-Del[4C4-C5]1Lap/Mmcd) mice, were obtained from Mutant Mouse Resource & Research Centre (MMRC, Davis, CA). In the mouse, ANRIL is located on Chromosome 4 within the *p15/CDKN2B-p16/CDKN2A-p14/ARF* gene cluster (INK4 locus) in an anti-sense manner. ANRILKO mice were generated on a 129S6/SvEv background where a 70kb region of the mouse genome corresponding to the human 58-kb noncoding RNA associated with coronary artery disease risk interval was deleted [15]. We have previously demonstrated lack of ANRIL expression in the ANRILKO mice tissues [12, 13].

A chemically-induced mouse model of Type 1 diabetic was generated with intraperitoneally injected streptozotocin (five doses, 50mg/kg in citrate buffer, pH5.6) on consecutive days [17, 18]. As our previous studies used male mice and to have a cost-containment, in this investigation we kept the same focus. We do however, understand the importance of using mice of both sexes and plan to carry out such analyses in the future. Age and sex-matched littermate controls received equal volume of citrate buffer. Diabetes was confirmed by blood glucose measurement (>20mmol/L) from tail vein using a glucometer. Animals were monitored for variations in body weight and blood glucose. Mice were sacrificed after 8 weeks of the onset of diabetes and tissues were collected. Renal cortices were dissected out from the kidneys and kept frozen at -80˚C for further analysis. A total of four groups of mice [n = 3/groups] were used, namely a) non-diabetic wild type control (WT-C), b) Wild type mice with diabetes (WT-D), c) ANRIL knockout control (KO-C) and d) ANRIL knockout mice with diabetes (KO-D).

### RNA sequencing

Three biological replicates were used for each of the four groups. Total RNA was extracted using the RNeasy Mini Kit (QIAGEN, Toronto, ON). The libraries were prepared by using the TruSeq RNA V2 kit. Sequencing was performed by Macrogen Corp. (Rockville, MD) on an Illumina sequencer with NovaSeq6000 S4 flow cell.

### Bioinformatics analysis of RNA-sequencing data

An average of 42 million (minimum of 27 million) paired-end reads were generated for each sample. Samples were aligned using STAR version 2.7.3a [19] to mouse genome mm10 with

GENCODE M23 primary annotations. An average of 30 million reads aligned uniquely (minimum 20 million). Subsequent analysis was performed using R version 3.6.1. Count matrices were generated using the summarized overlaps function from the Genomic Alignments package version 1.20.1 [20]. Differential expression analysis was performed using DESeq2 version 1.24.0 [21] and edgeR version 3.28.0 24 [22]. A gene was considered to be differentially expressed if it had an adjusted p-value < 0.01.

### KEGG and Reactome pathway analysis

Pathway analyses are extremely useful. Although, pathway analyses are usually performed by using a single pathway database, various available pathway databases are different in terms of the biochemical interactions and the pathway subcategories they use, Hence, to improve the predictions and augment the likelihood of identifying the relevant pathways accurately, we used two pathway analyses tools. After we identified differentially expressed genes, we applied KEGG [23–25] and Reactome [26] pathway analyses to identify pathways that are enriched for significant genes in (a) wild-type control vs. wild-type diabetic mice (WT-C vs. WT-D), and (b) wild-type control vs. diabetic knockout mice (WT-C vs. KO-D). In order to perform KEGG and Reactome pathway analyse, we chose to use the built-in functions available in STRING database and web resource (https://string-db.org). We identified a list of significant pathways with false discovery rate (FDR) < 0.01. We also measured the number of down-regulated and up-regulated genes from our list of genes that map to each pathway. Additionally, STRING provided a list of significant biological processes and molecular functions enriched for each comparison at the FDR < 0.01.

### Protein-protein interactions and cluster analysis

In addition to employing STRING functions to identify KEGG and Reactome pathways and the relevant biological processes and molecular functions, we used STRING to visualize protein-protein interaction (PPI) networks for each comparison (WT-C vs. WT-D, and WT-C vs. KO-D), and to identify clusters of interacting genes. Briefly, SRTING integrates both predicted and known PPIs to predict functional interactions of proteins, where each gene/gene product is presented by a node in the network, and the biological relationship between pairs of genes is represented by an edge between the two nodes.

We used STRING implementation of Markov Cluster Algorithm (MCL) with the default parameters to identify clusters of interacting genes (i.e. sub-networks) in each PPI network. We then considered each major cluster individually and identified their enriched pathways and molecular functions. This provided insight into the underlying molecular mechanisms at a finer resolution (i.e. at the cluster-level).

### IPA analysis

To further identify molecular sub-networks associated with each comparison (WT-C vs. WT-D and WT-C vs. KO-D), we used QIAGEN Ingenuity Pathway Analysis (QIAGEN IPA) software (QIAGEN Inc., https://www.qiagenbio-informatics.com/products/ingenuity-pathway-analysis), as an alternative network analysis approach. We uploaded a list of gene symbols along with their corresponding edgeR adjusted P values and log fold changes into the software, and set the analysis parameter to include genes with FDR < 0.01. Briefly, IPA uses the Ingenuity Pathways Knowledge Base to identify networks of the genes based on their connectivity and ranks networks based on the assigned scores. These scores take into the account the size of the network and the number of genes to predict if a network is relevant. Once the networks are ranked based on their assigned scores, they are presented by graphs that indicate

the molecular relationships between genes or gene products. Here genes are presented by nodes, and the biological relationships between pairs of genes are represented by edges.

## Results

### Diabetes induced metabolic alteration and renal damage are prevented by nullifying ANRIL production

Diabetic animals were maintained for a period of eight weeks following onset of hyperglycemia and were compared with Age- and sex- matched controls. Wild type diabetic (WT-D) animals showed hyperglycemia, reduced body weight, polyuria (urine volume >25ml/day in WT-D, compared to <3 ml /day in the littermate controls, WT-C) and glycosuria (not shown). These are characteristic metabolic features of poorly controlled type 1 diabetes. ANRIL KO diabetic mice (KO-D) remained hyperglycemic. However, they showed reduction of urine volume (<10ml/day) compared to WT-D mice. In keeping with previous analyses, we also calculated urinary albumin creatinine ratio (Fig 1). Urinary albumin/creatinine ratios, were elevated in WT-D mice compared to wild type controls (WT-C), as a result of renal damage and were normalized in the KO-D mice, indicating a protective effect of nullifying ANRIL.

### Diabetes-induced alterations of a large number of transcripts are prevented following ANRIL KO

We used two bioinformatics programs, (edgeR and DESeq2) to identify the genes that are differentially expressed between multiple groups of samples. Our aims were first to identify the differentially expressed transcripts in the kidneys in the context of DN by comparing wild-type control (WT-C) with wild-type diabetic (WT-D) mice. We then tried to identify the differentially expressed transcripts that are regulated through ANRIL by comparing wild-type non-diabetic controls (WT-C) and diabetic knockout (KO-D) tissues. We also examined any alterations of transcripts in the kidneys of non-diabetic animals caused nullification of ANRIL by comparing WT-C and KO-C. Overall alterations of the transcripts are graphically depicted in Fig 2. Differentially expressed genes detected by EdgeR are mostly a subset of those detected by DESeq2 (Fig 2A and 2D). Since EdgeR was more stringent and may generate less false positive results, it is possibly a better option compared to DESeq2. However, in both such analyses, minimal number of alterations of transcripts were seen at the basal level (WT-C vs KO-C), suggesting a minimal impact of these knockout on the downstream transcript at the basal level. On the other hand, poorly controlled Diabetes caused alterations of a large number of transcripts in the wild type (WT-D) animals compared to the wild type controls (WT-C), while several of those genes were not significantly different between ANRIL knockout diabetic mice (KO-D) and the wild type controls (WT-C). For example, in edgeR analyses, number of differentially expressed genes between WT-C and WT-D at p value < 0.05 was 4,153 and that at the p value <0.01 was 1,500. After ANRIL knockout 3,436 and 1,235 of these genes remained significant, for p values < 0.05 and 0.01, respectively (Fig 2). NGS data have been submitted in the public database (GEO ID: GSE197699).

### ANRIL regulates multiple DN related transcripts

We then sought to identify specific pathways that are altered in DN. We used KEGG and Reactome pathway analyses for those genes differentially expressed between WT-C and WT-D mice. Both KEGG and Reactome pathway analyses at P < 0.05 level showed a large number of alterations of transcripts (S1A and S2A Figs). Of specific relevance to DN (WT-C vs. WT-D; P < 0.01), these transcripts include metabolic pathways, apoptosis, extracellular matrix protein

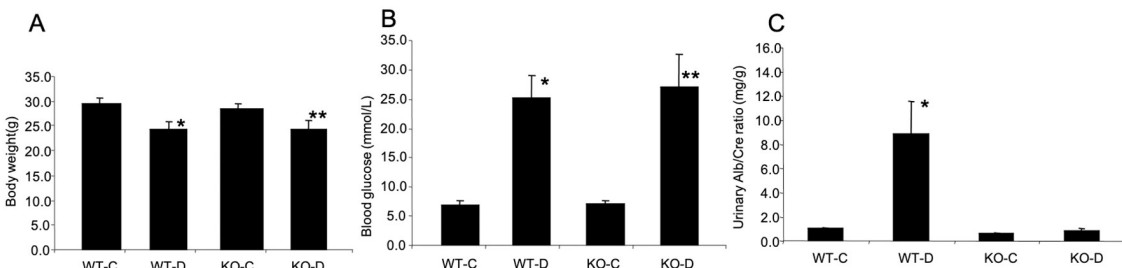

**Fig 1. Clinical monitoring.** Diabetes-induced A) reduced body weight, B) elevated blood glucose levels and C) increased urinary albumin/creatinine ratios were prevented in the diabetic animals lacking ANRIL (KO-D). (WT-C = wild type controls, WT-D wild-type diabetic, KO-C = ANRIL knockout controls, *P = 0.05 or less vs. WT-C, ** P = 0.05 or less vs. KO-C).

synthesis and degradation, NFKB related pathways, AGE-RAGE interaction pathways etc. (Fig 3A and 3E). The detailed listings of these pathways and transcripts are in the S1 Table (KEGG analysis; WT-C vs. WT-D) and in S2 Table (Reactome analysis; WT-C vs. WT-D). ANRIL KO prevented majority of these pathways in the diabetic animals. At the $P < 0.01$ level, a main group of transcripts which were not normalized in KEGG or Reactome analysis were mRNAs related to metabolic pathways in the diabetic animals (Fig 3C and 3G). Given that these

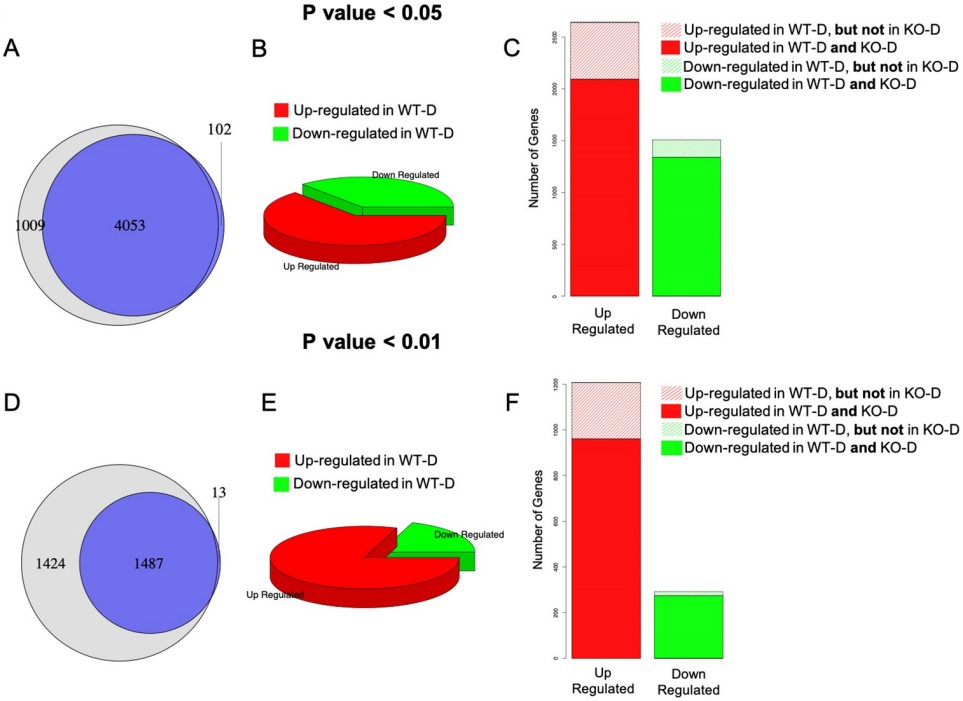

**Fig 2. Differentially expressed genes.** Differentially expressed genes identified by DESeq2 (light grey) and EdgeR (blue) [WT = wild-type, D = diabetes, C = age matched non-diabetic controls, KO = Knockout]. Data are presented at a p value threshold of 0.05 (A, B, C) and at a p value threshold of 0.01 (D, E, F). The differentially expressed genes (WT-C vs. WT-D) identified by EdgeR highly overlap those of DESeq2 (A, D). Poorly controlled diabetes caused alterations of a large number of transcripts in the wild type (WT) animals (A, D). Most of these differentially expressed transcripts are up-regulated in wild-type diabetic compared to wild-type controls (B, E). ANRIL knockout (KO) prevented several of such alterations. Several of the genes detected to be differentially expressed between WT-C and WT-D are not differentially expressed between WT-C and KO-D (C, F).

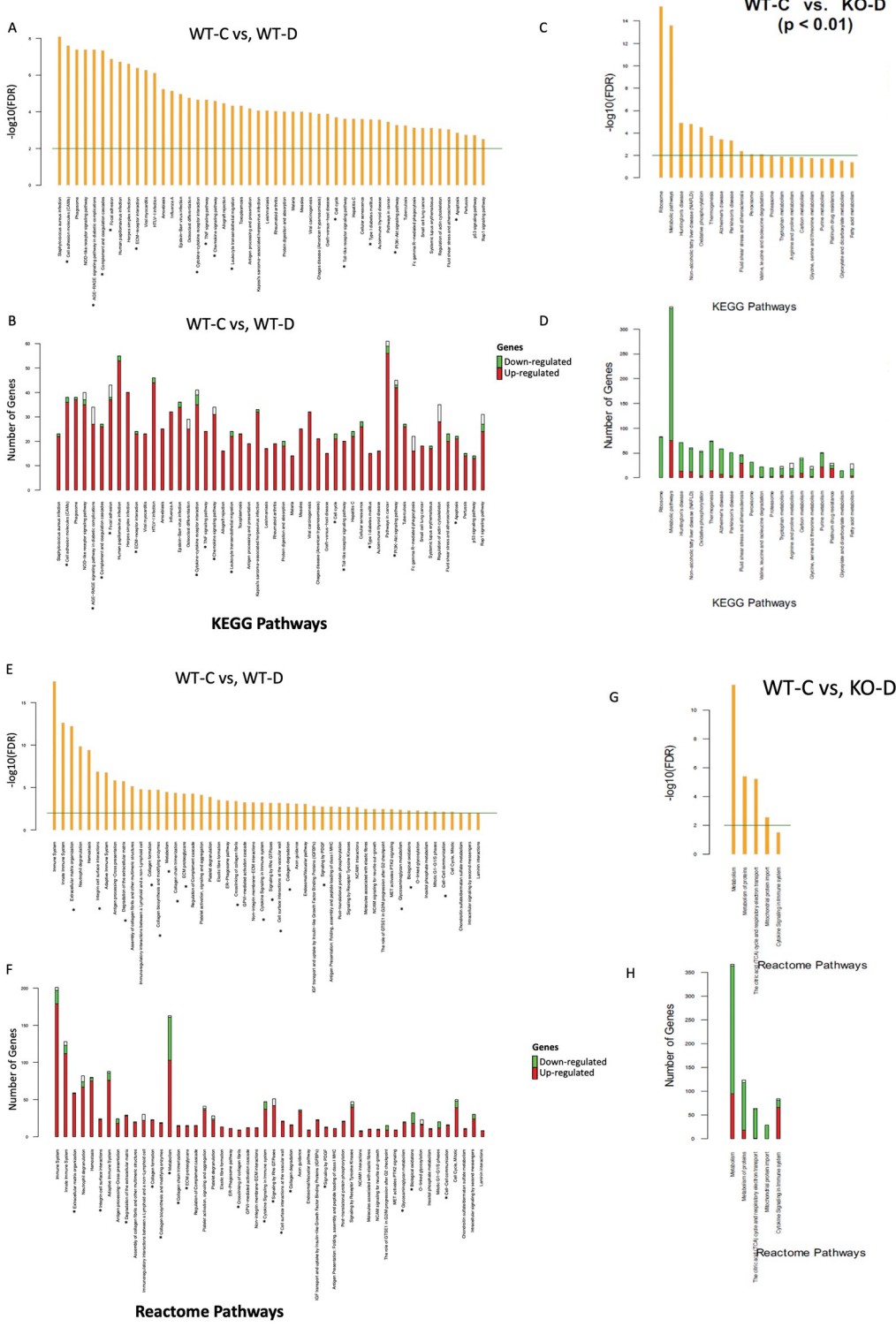

**Fig 3. Top-ranked pathways and number of significant genes from each pathway.** (A) Significant KEGG pathways enriched for genes differentially expressed (adjusted p < 0.01) between wild-type controls and wild-type diabetes mice (WT-C vs. WT-D), and (B) the number of down-regulated (green), up-regulated (red) and non-significant (white) genes mapped to each pathway. (C) KEGG pathway analysis showed that metabolic pathway is enriched for genes differentially expressed (adjusted p < 0.01) between wild-type controls and knockout diabetic mice (WT-C vs. KO-D), and (D) the number of down-regulated (green), up-regulated (red) and non-significant (white) genes mapped to each pathway. List of top Reactome pathways for WT-C vs. WT-D (E) and WT-C vs. KO-D (G), and the number of genes

mapped to each pathway (F, H). Down-regulated and up-regulated genes are shown in green and red, respectively. Both KEGG (A) and Reactome (E) pathway analyses showed alterations (p<0.01) of transcripts related to multiple biological pathways in the kidneys in diabetes. The majority of these, except for the transcripts related to metabolic pathways in KEGG analyses, were corrected in the ANRIL KO mice (C). Reactome analyses also confirmed similar patterns (G). (C = non-diabetic control, D = poorly controlled diabetic, WT = wild type, KO = ANRIL knockout, * = some pathways known to alter in diabetic kidney diseases, Horizontal line = FDR of 0.01, the analyses related to genes differentially expressed at the level of P<0.05 have been depicted in S1 Fig). The detailed listings of these transcripts are in the supplementary tables (S1 and S2 Tables).

animals are hyperglycemic, the finding the transcripts in metabolic pathways in pathway analysis that were not normalized is not surprising (Fig 3).

Overall, these data indicate that lncRNA ANRIL regulate multiple transcripts with pathogenetic role in DN. Furthermore, it is of particular interest that compared to the normal (WT) animal ANRIL KO mice without diabetes had no significant alterations with respect to mRNA expression at the basal level.

## Protein-protein interactions and cluster analysis

We performed protein-protein interaction (PPI) network analysis using STRING in order to uncover relationships between various differentially expressed transcripts in DN (adjusted p value < 0.01). We observed that the transcripts that were differentially significant between wild-type diabetes and wild-type controls (WT-C vs. WT-D) formed multiple sub-networks (Fig 4A). These finding suggest that there are extensive interaction of multiple molecules and pathways, which works in a co-ordinated fashion to produce diabetes induced changes in the kidneys.

We further undertook a cluster analysis using MCL to identify the major clusters of differentially expressed transcripts in our PPI network. Such analyses revealed that four major clusters (with 29, 31, 70 and 81 genes per cluster) existed in the PPI network of the differentially expressed genes in the renal tissues in DN. A diagrammatic representation of these clusters is shown in Fig 4A.

We reasoned that each cluster of highly interacting genes in WT-C vs WT-D PPI network (Fig 4A) may have their own distinct functions, that might be obscured when we consider all genes in the network together. We, therefore, identified KEGG and Reactome pathways, molecular functions and biological processes of each individual cluster (S3 Table). A detailed listing of the top KEGG pathways for each of the four major clusters is presented in Table 1. Some of the key genes and pathways altered in diabetes were identified in these clusters including collagens, TGFβ, PDGFR, PARPs, along with molecules which haven't been characterised yet in the context of DN.

Canonical pathway and sub-network analysis using IPA: We carried our further analyses using QIAGEN IPA software. Our analyses showed that a large number of transcripts in overlapping canonical pathways are altered in the kidneys in diabetes. Fig 4B depicts overlaps of enriched IPA canonical pathways for wild-type control vs. wild-type diabetic mice (WT-C vs. WT-D). As Fig 4B suggests the canonical pathways in WT-C vs. WT-D are highly overlapping. This implies that the altered transcripts are normally expressed in tissues, in diabetes, they are overexpressed causing subsequent tissue damage and development of DN.

We further used QIAGEN IPA software to identify the top-ranked networks, and their relevant molecules and top diseases and functions for WT-C vs. WT-D (Table 2). Fig 5 shows networks 3, 4 and 17 of Table 2. These networks correspond to TGFβ1, VEGF and collagen

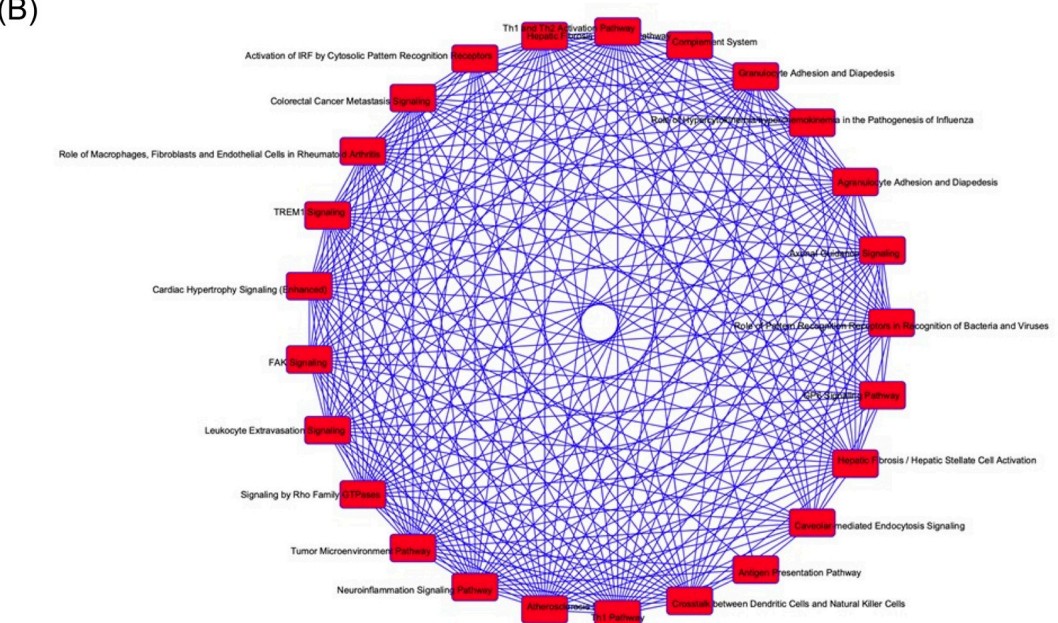

**Fig 4. Cluster analysis.** Cluster analyses showing when comparing (A) Wild-type diabetic with wild-type non-diabetic controls (WT-C vs. WT-D), the differentially expressed transcripts were organized in at least 4 major clusters (with > 25 genes within each cluster). However, when (B) Overlaps of enriched IPA canonical pathways. Interactions of enriched canonical pathways are shown for wild-type controls vs. diabetes (WT-C vs. WT-D). Each node corresponds to one canonical pathway, and the lines connecting pathways indicate interactions. Nodes (i.e. pathways) are colored based on their significance levels; such that darker nodes represent more significant pathways. Threshold of significance was defined as FDR < 0.01.

**Table 1. STRING protein-protein interaction networks for WT-C vs. WT-D.**

| A Cluster 1 (Red) | | | |
|---|---|---|---|
| KEGG Pathway | Observed Gene Count | False Discovery Rate (FDR) | Matching Proteins |
| Cell cycle | 16 | 2.60E-18 | Cdc20,Cdk1,Mcm4,Pkmyt1,Mcm6,Bub1,Ccnb2,Bub1b,Mcm3,Espl1,Ttk, Ccnb1,Cdc26,Cdc6,E2f1,Mcm5 |
| Oocyte meiosis | 9 | 1.55E-08 | Cdc20,Cdk1,Pkmyt1,Sgol1,Bub1,Ccnb2,Espl1,Ccnb1,Cdc26 |
| DNA replication | 6 | 1.37E-07 | Pole,Mcm4,Mcm6,Mcm3,Pole4,Mcm5 |
| Progesterone-mediated oocyte maturation | 6 | 1.81E-05 | Cdk1,Pkmyt1,Bub1,Ccnb2,Ccnb1,Cdc26 |
| p53 signaling pathway | 5 | 6.37E-05 | Cdk1,Ccng1,Ccnb2,Ccnb1,Gtse1 |
| HTLV-I infection | 7 | 0.00059 | Cdc20,Pole,Ccnb2,Bub1b,Cdc26,Pole4,E2f1 |
| MicroRNAs in cancer | 4 | 0.0134 | Brca1,Ccng1,Cdca5,E2f1 |
| Platinum drug resistance | 3 | 0.0195 | Brca1,Top2a,Birc5 |
| Cellular senescence | 4 | 0.0222 | Cdk1,Ccnb2,Ccnb1,E2f1 |
| Base excision repair | 2 | 0.0371 | Pole,Pole4 |
| Homologous recombination | 2 | 0.0459 | Brca1,Rad51 |

| B Cluster 2 (Aquamarine) | | | |
|---|---|---|---|
| KEGG Pathway | Observed Gene Count | False Discovery Rate (FDR) | Matching Proteins |
| Influenza A | 13 | 3.52E-13 | Rsad2,Eif2ak2,Irf7,Ifih1,Oas3,Ddx58,Cxcl10,Oas2,Oas1a,Oas1g,Stat2, Adar,Irf9 |
| Measles | 11 | 1.27E-11 | Eif2ak2,Irf7,Ifih1,Oas3,Ddx58,Oas2,Oas1a,Oas1g,Stat2,Adar,Irf9 |
| Herpes simplex infection | 12 | 3.10E-11 | Eif2ak2,Irf7,Ifih1,Oas3,Ddx58,Oas2,Sp100,Oas1a,Oas1g,Ifit1,Stat2,Irf9 |
| NOD-like receptor signaling pathway | 11 | 5.80E-11 | Irf7,Oas3,Gbp7,Oas2,Oas1a,Oas1g,Stat2,Gbp3,Ifi204,Tmem173,Irf9 |
| Hepatitis C | 10 | 1.53E-10 | Eif2ak2,Irf7,Oas3,Ddx58,Oas2,Oas1a,Oas1g,Ifit1,Stat2,Irf9 |
| RIG-I-like receptor signaling pathway | 7 | 1.89E-08 | Dhx58,Irf7,Ifih1,Ddx58,Cxcl10,Isg15,Tmem173 |
| Cytosolic DNA-sensing pathway | 6 | 0.000000287 | Irf7,Zbp1,Ddx58,Cxcl10,Adar,Tmem173 |
| Necroptosis | 5 | 0.00061 | Eif2ak2,Zbp1,Mlkl,Stat2,Irf9 |
| Human papillomavirus infection | 6 | 0.0024 | Eif2ak2,Oasl1,Oasl2,Isg15,Stat2,Irf9 |
| Hepatitis B | 4 | 0.0035 | Irf7,Ifih1,Ddx58,Stat2 |
| Kaposi's sarcoma-associated herpesvirus infection | 4 | 0.0104 | Eif2ak2,Irf7,Stat2,Irf9 |
| Viral carcinogenesis | 4 | 0.0104 | Eif2ak2,Irf7,Sp100,Irf9 |
| TNF signaling pathway | 3 | 0.0118 | Cxcl10,Mlkl,Ifi47 |

| C Cluster 3 (Sky Blue) | | | |
|---|---|---|---|
| KEGG Pathway | Observed Gene Count | False Discovery Rate (FDR) | Matching Proteins |
| Leishmaniasis | 5 | 4.28E-06 | Itgb2,Cybb,Itga4,Fcgr3,Ncf2 |
| Phagosome | 5 | 1.80E-04 | Itgb2,Cybb,Ctss,Fcgr3,Ncf2 |
| Tuberculosis | 5 | 1.80E-04 | Itgb2,Ctss,Il10ra,Fcer1g,Fcgr3 |
| Leukocyte transendothelial migration | 4 | 5.50E-04 | Itgb2,Cybb,Itga4,Ncf2 |
| Osteoclast differentiation | 3 | 1.38E-02 | Lilrb4,Fcgr3,Ncf2 |
| Cell adhesion molecules (CAMs) | 3 | 0.0238 | Itgb2,Cd86,Itga4 |
| Intestinal immune network for IgA production | 2 | 0.0238 | Cd86,Itga4 |
| Staphylococcus aureus infection | 2 | 0.0294 | Itgb2,Fcgr3 |

| D Cluster 4 (Cornflower Blue) | | | |
|---|---|---|---|
| KEGG Pathway | Observed Gene Count | False Discovery Rate (FDR) | Matching Proteins |
| Protein digestion and absorption | 8 | 7.61E-12 | Col6a1,Col6a2,Col1a1,Col5a1,Col1a2,Col4a1,Col4a2,Col5a2 |
| ECM-receptor interaction | 7 | 1.55E-10 | Col6a1,Col6a2,Col1a1,Col1a2,Col4a1,Col4a2,Hspg2 |

*(Continued)*

**Table 1.** (Continued)

| | | | |
|---|---|---|---|
| Focal adhesion | 6 | 1.19E-06 | Col6a1,Col6a2,Col1a1,Col1a2,Col4a1,Col4a2 |
| Human papillomavirus infection | 6 | 1.94E-05 | Col6a1,Col6a2,Col1a1,Col1a2,Col4a1,Col4a2 |
| PI3K-Akt signaling pathway | 6 | 1.96E-05 | Col6a1,Col6a2,Col1a1,Col1a2,Col4a1,Col4a2 |
| AGE-RAGE signaling pathway in diabetic complications | 4 | 0.0000286 | Col1a1,Col1a2,Col4a1,Col4a2 |
| Amoebiasis | 4 | 0.0000295 | Col1a1,Col1a2,Col4a1,Col4a2 |
| Relaxin signaling pathway | 4 | 0.0000582 | Col1a1,Col1a2,Col4a1,Col4a2 |
| Proteoglycans in cancer | 4 | 0.00026 | Col1a1,Col1a2,Dcn,Hspg2 |
| Small cell lung cancer | 2 | 0.0115 | Col4a1,Col4a2 |
| Platelet activation | 2 | 0.0178 | Col1a1,Col1a2 |

molecules. We selected these networks as these are well established molecules of pathogenetic significance in DN [1, 2, 5, 6].

## Discussion

At the transcription and post transcription levels, a symphony of regulatory molecules including transcription factors, transcription co-activators, DNA and histone methylators, non-coding RNAs and others have been shown to play critical roles in altered protein production [27]. Such epigenetic mechanisms play a major role in all chronic disease processes including DN [7, 8].

Long non-coding RNAs with a size of >200 bp play a major role in gene regulation [9]. Long non-coding RNAs, in general, acts through multiple mechanisms and may on a nearby [cis] or distant genes [trans] regulating their transcription [9, 10]. We took a novel approach using RNA sequencing and bioinformatics analysis to answer the role of a specific lncRNA, *ANRIL* in DN.

We have previously demonstrated a pathogenetic role of *ANRIL* on the pathogenesis of several chronic diabetic complications including DN [13]. The major aim of this study was to delineate the mechanisms of such protection.

In this study, using RNA sequencing and bioinformatic analyses approaches, we demonstrated that the vast majority of the altered renal transcripts, in the context of DN, are regulated by *ANRIL* as they were mostly normalized in the diabetic ANRIL KO mice. In the kidneys of wild type diabetic animals, as a result of diabetic dysmetabolism, RNA transcripts involved in a large number of pathways were changed. The major pathways, as observed in both KEGG and REACTOME analyses, include AGE-RAGE signaling, PI3K-AKT, metabolic pathways, TNFα and NF KB related molecules, multiple extracellular matrix proteins and growth factor related pathways were changed. These pathways are known to mediate renal damage in DN. In the kidneys of these animals we have earlier validated several of such molecules [12, 13]. Although in the current study, we didn't examine specific molecules, we have already performed such analysis using the same animal groups in the context of DN [13]. However, a large number of additional molecules, yet to be characterized in DN were also changed. In the kidneys of the diabetic KO animals, even in the presence of hyperglycemia, all important pathways were corrected and resulted in the amelioration of renal functional changes in DN. The exact significance of the remaining molecules, which were not corrected, are not clear. It is of further interest to note that as the animals remained hyperglycemic, the metabolic pathway related genes were not normalized in ANRIL KO mice with diabetes (KO-D).

**Table 2. Top-ranked sub-networks identified by QIAGEN IPA software for WT-C vs. WT-D.** Up-regulated and down-regulated molecules are shown with up and down arrows, respectively.

| Molecules in Network | Top Diseases and Functions |
|---|---|
| 1. ↑ ASPM, ↑ BRCA1, ↑CCNB1, ↑ CDCA5, ↑ CDK1, ↑ CENPF, ↑ CEP55, Cop9 Signalosome, ↑ CYTH3, ↑ DLGAP5 | Cell Cycle, Cellular Assembly and Organization |
| 2. ↑ DTX3L, ↑ GBP3, ↑ GBP5, ↑ GBP6, ↑ GBP7, ↑ HERC6, ↑ lfi27l2a/lfi27l2b, ↑IFI44, ↑ IFIT1B, ↑ IFIT2 | Antimicrobial Response, Immunological Disease |
| 3. Abl1/2, ↑ ADAMTSL5, ↑ ARHGAP19, ↓ CALML4, ↑ Clec2d (includes others), ↑ COL12A1, ↑ COL15A1, | Cancer, Connective Tissue Disorders Organ |
| 4. ↓ Akr1c14, ↓ ANGPTL7, ↑ ARHGAP23, ↑ ARHGEF17, ↑ BMPER, ↑ C1QTNF1, ↑ CDC42EP4, ↑ CHST1, ↑ CHST15 | Carbohydrate Metabolism, Connective Tissue |
| 5. 14-3-3, ↓ ACAT1, ↑ CYTH4, EGLN, ↑ EPAS1, ↑ ESYT1, CUFLNC, ↓ GUCD1, ↑HK2, ↑ ITGA4 | Carbohydrate Metabolism, Cardiovascular System |
| 6. ↑ ACSBG1, ↓ AKIP1, Akt, ↑ANGPTL2, ↑ ASTN2, ↑ ATP1B2, ↑ C1QC, ↑ C9orf116, ↑ CD93, ↑ CMTM7 | Carbohydrate Metabolism, Molecular Transport |
| 7. ↑ ALPK1, ↑ C1QTNF7, CD80/CD86, ↑ CLEC12a, ↑ CMPK2, ↑ Cxcl11, ↑ DDX58, ↑ DDX60, ↑EPSTI1 | Antimicrobial Response Immunological Disease |
| 8. ↑ ACTC1, ↑ ATAD2, ↑ ATP8B2, ATPase, Calmodulin, Cathepsin, ↑, CKAP2L, ↑ CTSC, ↓ CTSH, ↑ CTSK | Cellular Assembly and Organization |
| 9. ↑ ANXA6, ↓ AS3MT, calpain, ↑ CAPNS, ↑ CAPN6, Cyclin E, ↑DENND2A, ↑ DLG4, ↑ EDN1, Fgfr | Cell Morphology, Cell-To-Cell Signaling |
| 10. ↑ ACTA2, ↑ ACTN1, ↑ ANXA3, ↑ CAV1, ↑ CD44, Creb, ↑CSRP1, cytochrome-c oxidase, ↑ EFHC2, ↓ Gimap9 | Cardiovascular System Development |
| 11. ↑ ACAP2, ↑ ARHGAP28, ↑ BIRC3, CD3, ↑ CD300LD, ↑Ear2 (includes others), ↓ EBP, ↑ FIGNL1, ↑ FPR2, ↑ FTCD | Cell-To-Cell Signaling and Interaction |
| 12. ↑ AEN, ↑ ANLN, ↓ APOM, ↑ ARHGAP11A, ↓ ARSG, AURK, ↑ BCHE, ↑ BTG2, ↑ C1orf198, Caspase 3/7 | Cellular Assembly and Organization, Cellular Signaling |
| 13. ↑ CDKL5, ↑ CLU, ↑ DLG2, ↑ DLGAP4, ↑ DSCAML1, ↑ FNY, glutathione transferase, Glutathione-S_transferase, | Cell-To-Cell Signaling and Interaction |
| 14. ↑ ACSF2, ↓ ACSM5, Adaptor protein 1, adhesion molecule, Aldose Reductase, ↑ ALOX5, ↑ DOCK2, ↓ E2F5, Eif2, | Nervous System Development and Function |
| 15. ADRB, Alpha 1 antitrypsin, ↑ANKRD1, ↑ASNS, ↑ ATF3, ↑ C3, ↑ C5AR1, ↓ CA4, CaMKII, ↑ DDIT3 | Cell Death and Survival, Organismal Injury |
| 16. ↓ ABHD14A, ↑ ADAR, atypical protein kinase C, ↑ AXL, BCR (complex), ↑ CLEC6A, ↑ CSF1, ↑ EFHD1, ↑ EPB41L2, | Cellular Movement, Hematological System Defect |
| 17. ↑ ADAMTSL2, ↑ BMP1, ↑ CASP14, ↑ CCN4, ↑ COL1A2, ↑ COL5A2, ↑ COL6A1, ↑ COL6A2, collagen Collagen Alpha1 | Dermatological Diseases and Conditions |
| 18. ↑apyrase, ↑ CD200, ↓ CD320, ↑ CYBRD1, ↑ CYP1B1, ↑ CYP2S1, ↓ CYP4B1, ↑ DAAM, ↑ FAM234B, ↑Fcer1 | Developmental Disorder, Molecular Transport |
| 19. ↑ ADAM11, ↑ ADAM22, ↑ ADAMTS1, ↑ ADAMTS12, ↑ADAMTS14, ↑ ADAMTS2, ↑ ADAMTS5, ↑ ADAMTS7, ↑ART | Connective Tissue Disorders, Organismal Injury |
| 20. ALT, ↑ ARHGAP45, ↑ B3GNT7, ↑ CD72, ↑ CLEC9A, ↑ CORO1A, cytokine receptor, ↓ Gm1123, GOT, ↑ HELZ2 | Hematological System Development |

As noted above we have previously reported that the expression of ANRIL was upregulated in DN and in other chronic diabetic complications and that ANRIL may also play a pathogenetic role in other chronic diabetic complications such as diabetic retinopathy and cardiomyopathy [12, 13]. In the context of our current research, we have also shown that nullifying ANRIL production prevents several biochemical, functional and structural changes in DN [13]. As planned, the current study delineates molecular mechanisms of ANRIL mediated pathogenesis in DN. As noted, that at the basal levels very few transcripts were affected by ANRIL nullification. Exact reason for this finding is not clear. However, this finding provides an intriguing possibility that specific ANRIL blockade may have relatively less adverse effects. However, such notion needs to be explored using specific experiment.

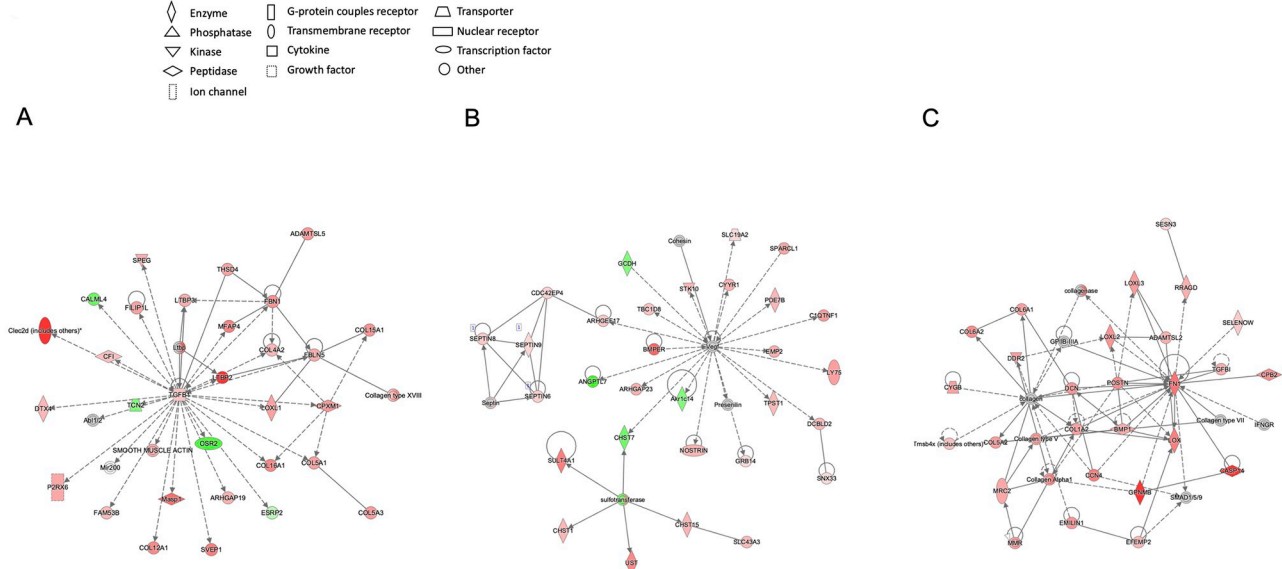

**Fig 5. Gene interaction network maps.** In the network, each gene or gene product is presented by a node, and the biological relationship between two nodes is presented by an edge between the two nodes. Color of the nodes indicate if they are down-regulated (green) or up-regulated (red). The darker colors show more significant genes and the lighter colors present less significant ones. Uncolored nodes represent genes that are not differentially expressed at a significant level in our dataset. However, they are presented in the network based on the evidence in the Ingenuity Pathways Knowledge Base, supporting their relationship to other genes in the network. The node shapes indicate their functions. The networks outlined here represent 3, 4 and 17 of Table 2, corresponding to A) TGFβ1, B) VEGF and C) collagen molecules. We selected these networks as these are well established molecules of pathogenetic significance in DN.

Although the role of ANRIL has been previously explored in the context of cardiovascular disease, very few studies have been conducted with respect to ANRIL in DN. We have demonstrated its role in regulating structural and function changes in DN. It has recently been demonstrated that ANRIL promotes pyroptosis and kidney injury in DN [28].

Exact mechanisms of ANRIL's action remains to be investigated. It has been postulated that ANRIL affects multiple genes by working through genome methylation via DNA methyltransferases and polycomb proteins, using both *cis* and *trans* mechanisms. Hence ANRIL may be a target for future anti-inflammatory drugs [29]. In other system ANRIL has been demonstrated to work by regulating microRNAs [30]. As microRNA alterations happen frequently in most disease conditions including diabetic complications, possibility that ANRIL works through specific microRNA remains a possibility and need to be further investigated [17, 18, 31, 32].

Based on our data it is possible that ANRIL may be considered as a drug target in DN and possibly other chronic diabetic complications. Based on these findings it is even tempting to speculate that ANRIL may potentially represent a one-stop shop for DN therapy. Such therapy may exploit an RNA based approach or a small molecule inhibitor targeting ANRIL. As no significant alterations of transcripts were seen in the non-diabetic ANRIL KO animals, such RNA targeting therapy conceptually may lead to limited side effects. However, such notion needs to be initially confirmed using long term large scale preclinical studies.

In summary, we have demonstrated, using an RNA sequencing and bioinformatics approach, that alterations of a large number of molecules associated with DN are regulated through lncRNA ANRIL. These finding suggest that ANRIL may potentially be a drug target for DN and possibly of other chronic diabetic complications:

## Supporting information

**S1 Fig. KEGG pathway analysis.** KEGG pathway analyses showed alterations (p<0.05) of transcripts related to multiple biological pathways in the kidneys in diabetes (A). The majority of these pathways were corrected in the ANRIL KO mice (B). (C = non-diabetic control, D = poorly controlled diabetic, WT = wild type, KO = ANRIL knockout. Horizontal line shows false discover rate of 0.01, the analyses for differentially expressed genes at the level of P<0.01 have been depicted in the Fig 3A. The detailed listings of these transcripts are in the supplementary tables (S1 and S2 Tables).
(TIF)

**S2 Fig. Reactome pathway analysis.** A) Reactome pathway analyses showed alterations (p<0.05) of transcripts related to multiple biological pathways in the kidneys in diabetes. (A). The majority of these pathways were corrected in the ANRIL KO mice (B). (C = non-diabetic control, D = poorly controlled diabetic, WT = wild type, KO = ANRIL knockout. Horizontal line shows false discover rate of 0.01, the analyses for differentially expressed genes at the level of P<0.01 have been depicted in the Fig 3E. The detailed listings of these transcripts are in the supplementary tables (S1 and S2 Tables).
(TIF)

**S1 Table. KEGG pathway analysis.**
(XLSX)

**S2 Table. Reactome pathway analysis.**
(XLSX)

**S3 Table. STRING analysis.**
(XLS)

## Author Contributions

**Conceptualization:** Subrata Chakrabarti.

**Formal analysis:** Parisa Sooshtari, Michael Levy, Hanxin Lin, Zhaoliang Su.

**Funding acquisition:** Subrata Chakrabarti.

**Investigation:** Biao Feng, Saumik Biswas.

**Methodology:** Biao Feng, Michael Levy, Hanxin Lin, Zhaoliang Su.

**Project administration:** Subrata Chakrabarti.

**Supervision:** Subrata Chakrabarti.

**Validation:** Parisa Sooshtari.

**Writing – original draft:** Parisa Sooshtari, Subrata Chakrabarti.

**Writing – review & editing:** Parisa Sooshtari, Biao Feng, Subrata Chakrabarti.

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
