## [Decision Letter · Decision Letter 0]

18 Jan 2022

PONE-D-21-38242ANRIL regulates multiple molecules of pathogenetic significance in diabetic nephropathyPLOS ONE

Dear Dr. Chakrabarti,

Thank you for submitting your manuscript to PLOS ONE. After careful consideration, we feel that it has merit but does not fully meet PLOS ONE’s publication criteria as it currently stands. Therefore, we invite you to submit a revised version of the manuscript that addresses the points raised during the review process. Please submit your revised manuscript by Mar 04 2022 11:59PM. If you will need more time than this to complete your revisions, please reply to this message or contact the journal office at plosone@plos.org. Please include the following items when submitting your revised manuscript:A rebuttal letter that responds to each point raised by the academic editor and reviewer(s). You should upload this letter as a separate file labeled 'Response to Reviewers'.A marked-up copy of your manuscript that highlights changes made to the original version. You should upload this as a separate file labeled 'Revised Manuscript with Track Changes'.An unmarked version of your revised paper without tracked changes. You should upload this as a separate file labeled 'Manuscript'.

We look forward to receiving your revised manuscript.

Kind regards,

Academic Editor

PLOS ONE

Journal Requirements:

[Supported by from the Canadian Institutes of Health Research (funding reference number: 169650) (SC), Schulich School of Medicine and Dentistry, Western University, Lawson Internal Research Fund (PS) and Jiangsu Province 100 talent International collaborative research program (BX2019100) (SC and ZS). PS is a recipient of new investigator award from Ontario Institute for Cancer Research (OICR).]

[The author(s) received no specific funding for this work.]

Reviewers' comments:

Reviewer's Responses to Questions

**Comments to the Author**

1. Is the manuscript technically sound, and do the data support the conclusions?

Reviewer #1: Yes

Reviewer #2: Partly

Reviewer #3: Yes

2. Has the statistical analysis been performed appropriately and rigorously? 

Reviewer #1: Yes

Reviewer #2: Yes

Reviewer #3: Yes

3. Have the authors made all data underlying the findings in their manuscript fully available?

Reviewer #1: Yes

Reviewer #2: Yes

Reviewer #3: Yes

4. Is the manuscript presented in an intelligible fashion and written in standard English?

Reviewer #1: Yes

Reviewer #2: Yes

Reviewer #3: Yes

5. Review Comments to the Author

Reviewer #1: In this study, Sooshtari et al. identified long non-coding RNA ANRIL, was upregulated, and plays a significant role in pathogenesis of diabetic nephropathy.

This study is original and interesting. The manuscript is well written, data are convincing, well presented. There are few areas needs to clarify,

1) Since there multiple non-coding RNAs involved in diabetic pathogenies, what is rational behind choosing lncRNA ANRIL?

2) Why author didn’t quantify Haemoglobin A1c (HbA1c) levels after mice induced with streptozotocin, did you find any difference in HA1c between WT-D vs KO-D cohort?

3) Does ANRIL-KO make any other complications in mice? Since, this KO alter multiple gene functions? And it plays functional role in pathogenesis of diabetic nephropathy

4) The study extensively did an in-silico approach for reporting functional regulation of genes, metabolic pathway alteration and protein interaction altered by ANRIL-KO, it would be more evidential, if it is confirmed with some real time experiments

5) What is reason behind using two different pathway analyzer KEGG and Reactome for pathway analysis?

6) Author identified an interesting factor that normal vs ANRIL-KO mice has no significant alterations in basal gene expression, what makes the significant difference in gene transcriptional level upon diabetic induced mice?

7) Did author generated any new data set from RNA sequencing between WT-D vs ANRIL-KO-D, if so then you didn’t submit to gene bank? If submitted what is the accession ID?

Limitation of the study is in-silico bioinformatic approach

Reviewer #2: Diabetes is the most common cause of ESRD, and thus molecular mechanisms in DN need to explore more to control the progression of DN. Epigenetic modification plays a crucial role in physiological and pathological conditions. The authors previously reported that lncRNA ANRIL arbitrated diabetes-associated abnormalities in DN using the ANRIL KO mice model. In this MS, the author addressed the downstream mechanisms of ANRIL alteration in DN based on RNA seq analysis using computational methods. They observed that ANRIL regulates multiple DN regulated transcripts. It is an expected observation, as the authors previously showed that ANRIL KO mice are protected from DN. Using protein-protein interaction and cluster analysis, they found a few non-characterized molecules, including known pathogenic proteins in the context of DN.

1. How many uncharacterized molecules are identified? Provide a list of molecules so that people can work in the future.

2. Does ANRIL regulate only DN-associated genes? How do you exclude other gens or uncharacterized genes in the context of DN?

3. ‘However, in both such analyses, minimal number of alterations of transcripts were seen at the basal level (WT-C vs KO-C), suggesting a minimal impact of these knockout on the downstream transcript at the basal level.’

Does ANRIL KO show any pre-programming by altering the DN-associated genes to protect diabetic-associated nephropathy? Are there have any differential expression genes in ANRIL KO compared to control (WT-C) in the context of DN?

4. Provide differentially expressed gene lists for each combination.

5. Does ANRIL regulate DN by targeting only DN-associated genes?

Minor revision.

1. The whole MS contains several track changes.

2. ‘Renal cortices were dissected out from the kidneys and kept frozen at 80oC for further’ It would be negative 80.

Overall, the authors analyzed RNA Seq data and used a bioinformatics approach to find the ANRIL mediated gene regulation. They did not prove any direct evidence that ANRIL modulates genes to protect DN. Still, this analysis would be a good source for future studies towards diabetes and diabetes-associated nephropathy.

Reviewer #3: Reviewer’s comment

Long non-coding RNA ANRIL is associated with coronary artery diseases, cancer, and diabetes. ANRIL is reported to have a role in diabetic kidney disease. In the study titled ‘ANRIL regulates multiple molecules of pathogenetic significance in diabetic nephropathy’ the authors Sooshtari et al have compared RNA expression in renal cortical tissues of ANRIL knockout (KO) mice and wild type (WT) mice, with or without streptozotocin (STZ) induced diabetes. Diabetic animals showed hyperglycemia, reduced body weight gain, polyuria, and increased urinary albumin. ANRIL knockout (KO) mice with and without STZ have corrected the polyuria and increased urinary albumin.

Differentially expressed genes were identified using edgeR and DESeq2. KEGG and Reactome pathway analyses and network analyses were done using STRING and IPA. Only a few genes were differentially expressed in control ANRIL knockout mice compared to control WT mice. Many genes were differentially expressed in Diabetic WT mice as compared to normal WT mice, and these genes belong to metabolic pathways, apoptosis, extracellular matrix protein synthesis and degradation, NFKB related pathways, AGE-RAGE interaction pathways, etc. ANRIL KO prevented most of these pathways.

This is an interesting study. The manuscript is well written, but the discussion part needs to be improved. Here are some suggestions which might improve the value of the article.

1. As per the manuscript, the study aims to identify the downstream mechanism of ANRIL in diabetic nephropathy. However, a detailed discussion on specific pathways of ANRIL is missing. Although the RNA expression varies with tissue type, if possible, the authors can find out the fate of known targets of ANRIL (Yin et al. 2021 PMID 33528317, Aarabi et al. 2018 PMID 29868613, Regmi et al. 2019 PMID 30835718, Ignarski et al. 2019 PMID 31277300, Dai et al. 2020 PMID 32256207) and include them in the discussion part too.

2. Check the table number and legends of Supplementary Table 2 and Supplementary Table 3

3. The authors have narrowed it down to some of the target genes of ANRIL. If possible, the authors can demonstrate changes in protein expression/phosphorylation of ANRIL targets in the tissues of KO mice or kidney cell lines by knocking down the ANRIL.

4. Is ANRIL regulates its targets through miRNA?

6. PLOS authors have the option to publish the peer review history of their article (what does this mean?). If published, this will include your full peer review and any attached files.

Reviewer #1: **Yes: **Purushoth Ethiraj

Reviewer #2: No

Reviewer #3: No

---

## [Author Response · Author response to Decision Letter 0]

9 May 2022

We sincerely thank the reviewers for their constructive comments. We believe that we have been able to adequately address all their comments and carried out necessary changes to address their comments. We believe that such changes have definitely improved the manuscript. Outline below are our specific responses:

Journal Requirements:

1. Please ensure that your manuscript meets PLOS ONE's style requirements, including those for file naming. …….

We checked to ensure that this manuscript follows the Journal style

2.We note that the grant information you provided in the ‘Funding Information’ and ‘Financial Disclosure’ sections do not match. 

We checked to ensure that these areas match

….. funding information should not appear in the Acknowledgments section or other areas of your manuscript. We will only publish funding information present in the Funding Statement section of the online submission form. 

 We changed as advised

 The data have been submitted, GEO ID: GSE197699. NCBI has confirmed that the data will be available as of March 6, 2022. We have provided GEO ID in the manuscript. The direct link if provided below: 

https://www.ncbi.nlm.nih.gov/geo/query/acc.cgi?acc=GSE197699

 

Reviewer 1:

Thank you for your encouraging and positive comments. Outlined below are our specific responses.

1) Since there multiple non-coding RNAs involved in diabetic pathogenies, what is rational behind choosing lncRNA ANRIL?

Selection of ANRIL was based on our previous findings. We have previously demonstrated that ANRIL is upregulated in several organs in diabetes (Refs. 12,13). Of further relevance to our current research, we have shown that nullifying the effects of ANRIL prevents several biochemical, functional and structural changes in DN (Ref 13). The current research is a direct extension of our previous study, to examine the mechanisms of ANRIL’s effect at the transcriptional levels. We have further explained this and added additional sentence in the introduction.

2) Why author didn’t quantify Haemoglobin A1c (HbA1c) levels after mice induced with streptozotocin, did you find any difference in HA1c between WT-D vs KO-D cohort?

This is a very important suggestion. We have carried out HbA1C earlier and found increased HbA1C in the STZ induced diabetic animals. As the KO-D mice showed high blood glucose levels along with lower body weight, we speculated that KO-D mice would have HbA1C levels similar to WT-D. However, this is an important question and we plan to do this in our future studies.

3) Does ANRIL-KO make any other complications in mice? Since, this KO alter multiple gene functions? And it plays functional role in pathogenesis of diabetic nephropathy

We haven’t observed any phenotypic alterations of ANRIL at the basal level (refs 12 and 13). However in the ANRIL KO mice diabetes induced cardiac and retinal changes are also prevented. Hence ANRIL may have a role beyond DN. We have added a comment regarding this in discussion.

 4) The study extensively did an in-silico approach for reporting functional regulation of genes, metabolic pathway alteration and protein interaction altered by ANRIL-KO, it would be more evidential, if it is confirmed with some real time experiments

This is an important suggestion. Outside of this manuscript, actually, we have already taken the suggested approach to confirm molecular alterations of specific pathway using the same animal model (ref 13). Such examinations were performed at mRNA (RT-PCR) and at protein level and include assessment of ECM proteins, growth factors, inflammatory markers. We however do recognize importance of such analyses and commented in discussion in this revised version.

5) What is reason behind using two different pathway analyzer KEGG and Reactome for pathway analysis?

Pathway analyses are extremely useful for the interpretation of genomics data, and therefore, a comprehensive set of databases, including KEGG and Reactome among others, have been generated in the past two decades for this purpose. Traditionally, pathway analyses are usually performed by using a single pathway database. However, different pathway databases are different in terms of the biochemical interactions they use, and the pathway subcategories. This has become a motivation for many researchers to use multiple pathway databases to improve their predictions, and increase the likelihood of identifying the relevant pathways accurately. We have inserted a comment regarding this.

6) Author identified an interesting factor that normal vs ANRIL-KO mice has no significant alterations in basal gene expression, what makes the significant difference in gene transcriptional level upon diabetic induced mice?

This is also a very interesting question. We however, don’t have a specific answer. It is possible that diabetic dysmetabolism alter the transcriptional machinery at a different level (sensitivity etc). On the positive side, it is potentially possible that if a drug, specifically targeting ANRIL is developed, it may have less adverse effects. However, these notions need to be established by specific experiments. We have inserted a comment in the discussion section regarding this observation.

7) Did author generated any new data set from RNA sequencing between WT-D vs ANRIL-KO-D, if so then you didn’t submit to gene bank? If submitted what is the accession ID?

Limitation of the study is in-silico bioinformatic approach

We had previously notified the Journal Editor that upon acceptance of this manuscript we will submit all data to a public database. The data has now been submitted. GEO ID: GSE197699 (https://www.ncbi.nlm.nih.gov/geo/query/acc.cgi?acc=GSE197699) and will be available from March 6, 2022

Reviewer 2:

Thank you for your encouraging and positive comments. Outlined below are our specific responses.

1. How many uncharacterized molecules are identified? Provide a list of molecules so that people can work in the future.

We have notified the Journal Editor that upon acceptance of this manuscript we will submit all data to a public database and accordingly provide such ID number. Such database will provide details of all molecules. The data has now been submitted. GEO ID: GSE197699 (https://www.ncbi.nlm.nih.gov/geo/query/acc.cgi?acc=GSE197699) and will be available from March 6, 2022

2. Does ANRIL regulate only DN-associated genes? How do you exclude other gens or uncharacterized genes in the context of DN?

Thank you for this thoughtful comment. Based our data and our previous publication in this area (Refs12, 13), it appears that ANRIL along with DN related genes, regulate multiple other genes. We have shown a protective effect of ANRIL in diabetic retinopathy and cardiomyopathy (refs, 12, 13). Other studies have demonstrated it diverse role in other conditions (refs. 10,11). The intricate and complex regulation of gene regulation by ANRIL and other lncRNAs have just began to unravel. Hence, it is very difficult to exclude other transcripts. In this research, we focused on DN related genes. Even in this study the identified transcripts may also have additional effects on other systems. Future research specifically targeting the altered transcript (list provided in the suppl. Tables) may be of help.

3. ‘However, in both such analyses, minimal number of alterations of transcripts were seen at the basal level (WT-C vs KO-C), suggesting a minimal impact of these knockout on the downstream transcript at the basal level.’

Does ANRIL KO show any pre-programming by altering the DN-associated genes to protect diabetic-associated nephropathy? Are there have any differential expression genes in ANRIL KO compared to control (WT-C) in the context of DN? 

In this revised version we have included the list of the genes altered in KO-C compared to WT-C. None of these are known DN related transcript and we feel that the changes seen in KO-D are possibly not due to pre-programming. Our differential gene expression results for WT-C vs. KO-C have been included in the GEO submission and will be publicly available from March 6, 2022. GEO ID: GSE197699 (https://www.ncbi.nlm.nih.gov/geo/query/acc.cgi?acc=GSE197699) 

4. Provide differentially expressed gene lists for each combination.

Some of these combinations have already been provided in the table and figures (including supplementary files). We have notified the Journal Editor that upon acceptance of this manuscript we will submit all data to a public database and accordingly provide such ID number. Such database will provide details of all molecules. Our differential gene expression results for WT-C vs. KO-D have been included in the GEO submission and will be publicly available from March 6, 2022. GEO ID: GSE197699 (https://www.ncbi.nlm.nih.gov/geo/query/acc.cgi?acc=GSE197699) 

5. Does ANRIL regulate DN by targeting only DN-associated genes?

Please see #2 above

Minor revision.

1. The whole MS contains several track changes

 Apologies, corrected in this revised version

2. ‘Renal cortices were dissected out from the kidneys and kept frozen at 80oC for further’ It would be negative 80.

Apologies, corrected in this revised version

Overall, the authors analyzed RNA Seq data and used a bioinformatics approach to find the ANRIL mediated gene regulation. They did not prove any direct evidence that ANRIL modulates genes to protect DN. Still, this analysis would be a good source for future studies towards diabetes and diabetes-associated nephropathy.

Thank you for this valuable comment

Reviewer 3:

Thank you for your encouraging and positive comments. Outlined below are our specific responses.

1. As per the manuscript, the study aims to identify the downstream mechanism of ANRIL in diabetic nephropathy. However, a detailed discussion on specific pathways of ANRIL is missing. Although the RNA expression varies with tissue type, if possible, the authors can find out the fate of known targets of ANRIL (Yin et al. 2021 PMID 33528317, Aarabi et al. 2018 PMID 29868613, Regmi et al. 2019 PMID 30835718, Ignarski et al. 2019 PMID 31277300, Dai et al. 2020 PMID 32256207) and include them in the discussion part too.

Thank you for this comment. In the revised version we have specifically addressed these researches in discussion and added the suggested references (refs. 29,30,31,32).

2.. Check the table number and legends of Supplementary Table 2 and Supplementary Table 3

Apologies, corrected in this revised version.

3. The authors have narrowed it down to some of the target genes of ANRIL. If possible, the authors can demonstrate changes in protein expression/phosphorylation of ANRIL targets in the tissues of KO mice or kidney cell lines by knocking down the ANRIL.

This is an important suggestion. Outside of this manuscript, actually, we have already taken the suggested approach to confirm molecular alterations of specific pathway using the same animal model (ref 13). Such examinations were performed at mRNA (RT-PCR) and at protein level and include assessment of ECM proteins, growth factors, inflammatory markers. We however do recognize importance of such analyses and commented in discussion in this revised version.

4. Is ANRIL regulates its targets through miRNA?

 This is a very thoughtful comment. It is possible that ANRIL and other lncRNAs work through microRNA. We have included a comment in the discussion outlining such possibility and added the suggested reference above.

---

## [Editor Report · Decision Letter 1]

8 Jun 2022

ANRIL regulates multiple molecules of pathogenetic significance in diabetic nephropathy

PONE-D-21-38242R1

Dear Dr. Chakrabarti,

We’re pleased to inform you that your manuscript has been judged scientifically suitable for publication and will be formally accepted for publication once it meets all outstanding technical requirements.

Kind regards,

Academic Editor

PLOS ONE
---

## [Editor Report · Acceptance letter]

13 Jun 2022

PONE-D-21-38242R1 

ANRIL regulates multiple molecules of pathogenetic significance in diabetic nephropathy 

Dear Dr. Chakrabarti:

I'm pleased to inform you that your manuscript has been deemed suitable for publication in PLOS ONE. Congratulations! Your manuscript is now with our production department. 

Kind regards, 

on behalf of

Dr. Rajakumar Anbazhagan 

Academic Editor

PLOS ONE